# Craniocervical Morphometry in Pomeranians—Part II: Associations with Chiari-like Malformation and Syringomyelia

**DOI:** 10.3390/ani14131859

**Published:** 2024-06-23

**Authors:** Koen Santifort, Sophie Bellekom, Ines Carrera, Paul Mandigers

**Affiliations:** 1IVC Evidensia Referral Hospital Arnhem, 6825 MB Arnhem, The Netherlands; 2IVC Evidensia Referral Hospital Hart van Brabant, 5144 AM Waalwijk, The Netherlands; 3Expertise Centre of Genetics, Department of Clinical Sciences, Faculty of Veterinary Medicine, Utrecht University, 3584 CS Utrecht, The Netherlands; 4Vet Oracle Teleradiology, Norfolk IP22 4ER, UK

**Keywords:** caudal cranial fossa, magnetic resonance imaging, computed tomography, clivus length, morphometry

## Abstract

**Simple Summary:**

Numerous diagnostic-imaging-based studies (computed tomography (CT) and magnetic resonance imaging (MRI)) have focused on the shape and size of the skull and neck of small- and toy-breed dogs in relation to disorders of the skull, vertebral column, and spinal cord. No such studies have particularly focused on associations with Chiari-like malformation and syringomyelia (CM/SM) in the Pomeranian dog breed. The aim Part II of this two-part study was to describe and analyze various aspects and measurements of the skull and neck region of Pomeranians in relationship to CM/SM by means of CT and MRI. Classifications of CM/SM and measurements on CT and MRI images were performed for 99 Pomeranians. Among other significant findings, we found that dogs with SM had a shorter skull base based on both MRI and CT measurements, and a significantly smaller skull in the area of the hindbrain. These results provide further insights into the development of CM/SM in Pomeranians.

**Abstract:**

Background: The aim of Part II of this two-part study is to describe and analyze the association of various aspects and measurements related to the morphometry of the skull and craniocervical region to CM/SM status of Pomeranians, by means of computed tomography (CT) and magnetic resonance imaging (MRI). Methods: Prospectively, Pomeranians were included that underwent both CT and MRI studies of the head and cervicothoracic vertebral column. For those cases where qualitative classifications differed between observers, the experienced observer re-evaluated the studies and decided on a final classification that was used for further analysis. For quantitative measurements, the means of the observers’ measurements were used for analysis. Results: Among statistically significant differences in measurements, we found that dogs with SM had a significantly shorter clivus length based on both MRI (*p* = 0.01) and CT measurements (*p* = 0.01), and a significantly smaller caudal cranial fossa area based on both MRI (*p* = 0.02) and CT measurements (*p* = 0.02). Conclusions: Significant morphometrical differences were identified between dogs with or without CM/SM. The findings in this study add to those already described in other breeds and provide further insight into factors that may play a role in the pathogenesis of CM/SM in Pomeranians.

## 1. Introduction

Chiari-like malformation (CM) and syringomyelia (SM) are disorders that have been documented in various dog breeds, including the Cavalier King Charles Spaniel (CKCS), Griffon Bruxellois, and other small- and toy-breed dogs [1,2,3,4,5]. In the CKCS, many imaging features, mostly based on magnetic resonance imaging (MRI), have been linked to CM/SM and have provided insights into pathophysiology as well as clinically relevant characteristics that can be communicated to owners and breeders. A review covers a variety of imaging features that have been studied in relationship to CM/SM [6].

A fairly large number of diagnostic-imaging-based (computed tomography (CT) and MRI) studies have focused on the morphology of the skull and craniocervical region of the CKCS and other mostly brachycephalic and small or toy breed dogs in relation to disorders such as atlantoaxial instability, atlantoaxial overlapping, and CM/SM [1,2,3,4,5,6,7,8,9,10,11,12,13,14,15,16,17]. These studies have provided valuable information on the pathogenesis of and relationships between these disorders.

Recently, we reported the prevalence of CM/SM, owner-reported clinical signs, and the associations thereof with a CM/SM grading scheme in Pomeranians [18]. However, morphological studies specifically focusing on the Pomeranian dog breed that evaluate the relationship of anatomical (imaging-based) features with CM/SM status are lacking. In Part I of this two-part study, we reported the intra-observer, interobserver, and intermodality (CT vs. MRI) reliability and agreement for various aspects of the skull and craniocervical region of Pomeranians with and without CM/SM, by means of CT and MRI. 

The aim of Part II of this study is to describe and analyze the association of various aspects and measurements related to the morphometry of the skull and craniocervical region to CM/SM status of Pomeranians by means of CT and MRI. Specifically, we aim to analyze for significant associations between CM/SM classification and CT measurements or MRI measurements of the cranial fossa and craniocervical junction. 

## 2. Materials and Methods

For this prospective study, Pomeranians were included that underwent both CT and MRI studies of the head and cervicothoracic vertebral column at two institutions (IVC Evidensia Small Animal Hospital Arnhem and IVC Evidensia Small Animal Hospital Hart van Brabant) during the period of February 2022 to October 2023. All owners agreed to participate in this study and informed consent was obtained. The study was approved by the Animal Welfare Body Utrecht, Utrecht University, The Netherlands. The studies were conducted in accordance with the local legislation and institutional requirements. 

Exclusion criteria included dogs with a prior history of or that were diagnosed with a central nervous system (CNS) disease on MRI other than one or more of the following (i.e., dogs affected by the following disorders were included): CM/SM, ventriculomegaly, supracollicular fluid accumulation, findings related to craniocervical junction abnormalities (CJA) (e.g., atlanto-occipital overlapping (AOO), dorsal constriction at C1/C2 (atlantoaxial band (AAB)), atlantoaxial instability (AAI, also referred to as atlantoaxial subluxation), or non-structural disorders such as epilepsy or paroxysmal dyskinesia. Dogs with MRI or CT studies with artifacts or insufficient image quality that did not allow for accurate assessments or measurements were also excluded.

MRI and CT studies were performed under general anesthesia (individualized anesthetic protocols) with a high-field MRI scanner (1.5T Canon Vantage Elan, The Netherlands) and 16-slice CT scanner (Siemens SOMATOM.go, The Netherlands). Dogs were positioned in sternal recumbency on the horizontal surface of the table with the head in a flexible coil (MRI) or a head rest (CT), both resulting in elevation of the head of around 2–3 cm above the table (Figure 1). MRI sequences obtained included sagittal T2W (echo time (TE) 110 ms, repetition time (TR) 2.6 s, 2.5 mm slice thickness, and 256 × 320 matrix), sagittal T1W (TE 10 ms, TR 0.5 s, 2.5 mm slices, and 256 × 320 matrix), transverse T2W of the brain (TE 115 ms, TR 4.1 s, 3.0 mm slices, and 160 × 192 matrix), transverse T2W of the cervical spinal cord (TE 115 ms, TR 4.1 s, 3.0 mm slices, and 160 × 192 matrix) and transverse T1W of the cervical spinal cord (TE 10 ms, TR 0.4 s, 3.0 mm slices, and 160 × 192 matrix). Transverse slices at the level of the cervical spinal cord were adjusted to center the syrinx, if visible. In dogs without a visible syrinx on sagittal images, transverse images were acquired at the level of C2-C3 vertebrae. CT scans were performed with the following parameters: 130 kVp tube voltage, 80 and 220 mAs tube current, 256 × 256 image matrix, 0.6 and 0.8 mm slice thickness, 0.4 and 0.6 mm slice increment, 1.0 s rotation time, and a pitch of 0.6. A bone algorithm was used for image reconstruction in transverse, dorsal, and sagittal planes. Three-dimensional reconstructions including soft tissues were obtained by use of imaging software (RadiAnt DICOM Viewer [Software version 2023.1]).

### 2.1. CT and MRI Evaluation

CM and SM classification and measurements were performed as described in the following paragraphs by multiple observers as described in Part I of this study. For those cases where qualitative classifications differed, the experienced observer re-evaluated the studies and decided on a final classification, which was used for further analysis. For quantitative measurements, the means of the measurements performed by the observers were used for analysis. All sequences and reconstructions were available to observers for evaluation.

#### 2.1.1. Classification of Chiari-like Malformation (CM)

Images were evaluated to assess the presence or absence of CM by evaluating the shape of the cerebellum and position of the caudoventral cerebellum (uvula). CM was classified as described previously [18], where CM normal = CM0 and CM abnormal = CM1 and CM2.

The line of the foramen magnum was defined as a straight line between the most ventral aspect of the supraoccipital bone and the most caudal aspect of the basioccipital bone on sagittal MR images (Figure 2).

#### 2.1.2. Classification of Syringomyelia (SM)

Images were evaluated to assess the presence or absence of SM in the spinal cord. SM was defined as a well-demarcated intramedullary lesion (or lesions) associated with the central canal of the spinal cord, hyperintense on T2W and hypointense on T1W images. SM was classified as previously described [18], where SM normal = SM0 and SM abnormal = SM1 and SM2.

Additionally, syrinx (when present) location was noted as follows: cervical, thoracic, extensive (both cervical and thoracic, continuous), or multifocal (both cervical and thoracic, discontinuous).

#### 2.1.3. MRI-Based Qualitative Parameters

The following additional qualitative parameters were assessed on midsagittal T2-weighted MR images:Presence of cerebrospinal fluid (CSF) signal between the cerebellum and the supraoccipitum (yes/no);Presence of CSF signal at the ventral aspect of the cervicomedullary junction (yes/no);Presence of CSF signal at the dorsal aspect of the cervicomedullary junction (yes/no).

#### 2.1.4. Quantitative MRI- and CT-Based Measurements 

Skull and vertebral quantitative morphometric measurements, performed on midsagittal CT reconstructions and midsagittal MR images included the following.

Cranial fossa (Figure 2): Distance between the os tentorium cerebelli and the dorsum sellae (yellow line);Length of the clivus (dorsum sella turcica to ventral margin of foramen magnum) (green line);Height of the foramen magnum (‘foramen magnum line’, red line);Distance between the cranial tip of dorsal arch of the atlas and the foramen magnum line (blue line)—when negative, these dogs were interpreted to have atlanto-occipital overlapping;Area between the yellow line and the red line and osseous structures (caudal cranial fossa area, Area 1);Area rostral to the yellow line (rostral and middle cranial fossa area, Area 2).

Craniocervical junction (Figure 3):Angle between the line from caudal tip of the basioccipital bone to the cranial tip of the dens axis (red line) and the line from the ventral aspect of the supraoccipital bone to the cranial tip of the dens axis (green line) (Angle 1);Angle between the line from caudal tip of the basioccipital bone to the cranial tip of the dens axis (red line) and the line from caudal tip of the basioccipital bone to the midpoint of the caudal endplate of the axis (blue line) (Angle 2).

**Figure 2 animals-14-01859-f002:**
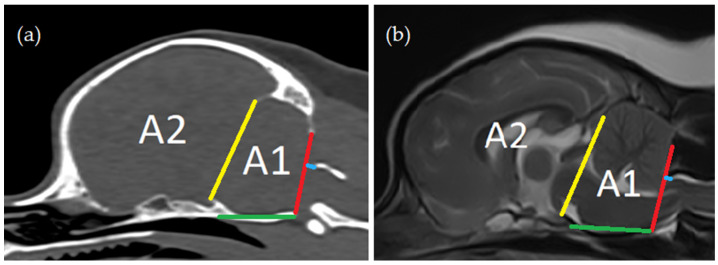
Cranial fossa measurements: (**a**) computed tomography, midsagittal plane reconstruction (bone window); (**b**) magnetic resonance imaging—T2-weighted midsagittal plane. Distance between the os tentorium cerebelli and the dorsum sellae (yellow line); length of the clivus (green line); height of the foramen magnum (‘foramen magnum line’, red line); distance between cranial tip of dorsal arch of the atlas and the foramen magnum line (blue line); area between the yellow line and the red line and osseous structures (caudal cranial fossa area, Area 1); and area rostral to the yellow line (rostral and middle cranial fossa area, Area 2).

**Figure 3 animals-14-01859-f003:**
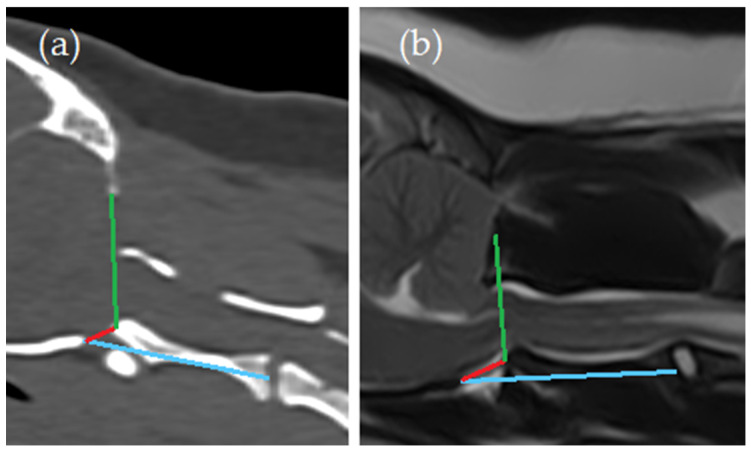
Craniocervical junction measurements. (**a**) Computed tomography, midsagittal plane reconstruction (bone window); (**b**) magnetic resonance imaging—T2-weighted midsagittal plane. Angle between the line from caudal tip of the basioccipital bone to the cranial tip of the dens axis (red line) and the line from the ventral aspect of the supraoccipital bone to the cranial tip of the dens axis (green line) (Angle 1); angle between the line from caudal tip of the basioccipital bone to the cranial tip of the dens axis (red line) and the line from caudal tip of the basioccipital bone to the midpoint of the caudal endplate of the axis (blue line) (Angle 2).

### 2.2. Statistical Analysis

Descriptive statistics are reported. Data were tested for normality using a Kolmogorov–Smirnov test.

Odds ratios were calculated for SM classification groups normal vs. abnormal for qualitative assessments. Odds ratios are reported with a 95% confidence interval (OR, 95% CI). Two-sample *t*-tests assuming unequal variances were performed to analyze for differences between CM and SM normal vs. abnormal group means of CT measurements and MRI measurements. *p*-values < 0.05 were regarded as significant. Statistical analyses were performed by making use of Microsoft Excel^®^ and R v4.3.1.

## 3. Results

### 3.1. Study Population

A total of 112 dogs were eligible for inclusion in the study, of which 13 were subsequently excluded. Ninety-nine (99) dogs were therefore included in the study. Table 1 includes characteristics of the study population. 

#### CM and SM Prevalence

The prevalences of classifications of CM and SM and classification as CM or SM normal vs. abnormal are included in contingency tables (Table 2 and Appendix A). Of the 99 dogs, 36 (36%) were classified as CM normal (CM0) and 63 (64%) were classified as CM abnormal (CM1 and CM2). Of the 63 CM abnormal dogs, 58 (92%; 59% of total) were classified as CM1 and 5 (8%; 5% of total) were classified as CM2.

Of the 99 dogs, 49 (49%) were classified as SM normal (SM0) and 50 (51%) were classified as SM abnormal (SM1 and SM2). Of the 50 SM abnormal dogs, 32 (64%; 32% of total) were classified as SM1 and 18 (36%, 18% of total) were classified as SM2.

Age and weight were not significantly different between CM normal (age: 2.7 (SD 1.6) years and weight: 3.4 (SD 1.0) kg) vs. CM abnormal dogs (age: 3.1 (SD 1.4) years and weight: 3.2 (SD 1.0) kg) or SM normal (age: 2.8 (SD 1.7) years and weight: 3.4 (SD 1.0)) vs. SM abnormal dogs (age: 3.0 (SD 1.4) years and weight: 3.3 (SD 0.9) kg) (*p* = 0.19–0.59).

### 3.2. Odds Ratios for Qualitative MRI Assessments of SM Classification Groups (Normal vs. Abnormal) 

The odds ratios (CI 95%) for qualitative MRI assessments of CM and SM normal vs. abnormal classification groups are included in Table 3 and Table 4, respectively. 

Dogs with or without CM and with or without SM were equally likely to show the presence of a CSF signal on MRI at the sites between the cerebellum and the supraoccipitum, the ventral aspect of the cervicomedullary junction, and the dorsal aspect of the cervicomedullary junction.

### 3.3. Differences between CM and SM Classification Groups (Normal vs. Abnormal) for MRI- and CT-Based Measurements

The means, standard deviations, and *p*-values of CM and SM normal vs. abnormal classification groups for MRI- and CT-based measurements are included in Table 5 and Table 6. 

#### 3.3.1. Differences between CM Classification Groups (Normal vs. Abnormal) for MRI- and CT-Based Measurements

Compared to CM normal dogs, CM abnormal dogs had:-A significantly larger foramen magnum (i.e., dorsoventral height on midsagittal images) based on MRI measurements (*p* = 0.02), but not based on CT measurements (*p* = 0.75);-A significantly greater distance between the cranial tip of dorsal arch of the atlas and the foramen magnum line based on MRI measurements (*p* = 0.01), but not based on CT measurements (*p* = 0.62).

#### 3.3.2. Differences between SM Classification Groups (Normal vs. Abnormal) for MRI- and CT-Based Measurements

Compared to SM normal dogs, SM abnormal dogs had:-A significantly shorter clivus length based on both MRI (*p* = 0.01) and CT measurements (*p* = 0.01);-A significantly larger foramen magnum (i.e., dorsoventral height on midsagittal images) based on MRI measurements (*p* = 0.01), but not based on CT measurements (*p* = 0.91);-A significantly shorter distance between the cranial tip of dorsal arch of the atlas and the foramen magnum line based on CT measurements (*p* = 0.04), but not based on MRI measurements (*p* = 0.92);-A significantly smaller caudal cranial fossa area (Area 1) based on both MRI (*p* = 0.02) and CT measurements (*p* = 0.02);-A significantly smaller rostral and middle cranial fossa area (Area 2) based on CT measurements (*p* = 0.046), but not based on MRI measurements (*p* = 0.20);-A significantly greater Angle 2 based on MRI measurements (*p* = 0.02), but not based on CT measurements (*p* = 0.60).

## 4. Discussion

This two-part study is the first large study specifically focusing on the Pomeranian dog breed that evaluates the relationship of various imaging-based anatomical features with CM/SM classification. The following paragraphs discuss the results of this study in comparison to previously published studies.

### 4.1. CM and SM Prevalence

The prevalence of CM (64%) reported in this study is slightly higher than the range of 53.4–62.4% that was previously reported for Pomeranians [18]. The prevalence of SM (51%) in this study was also higher than previously reported (11.9–34.7%). The reason for this observation is not entirely clear, but since the previous study included a more than 7-fold larger study population, this is most likely related to sampling bias. Alternatively, it may be that more affected dogs were presented for imaging studies over the last years relative to unaffected dogs (i.e., fewer pure screening imaging studies). We postulate that this might be due to increased awareness of owners regarding clinical signs exhibited by affected dogs [18]. The role of social media in sharing information and phenomenology of these signs between owners likely catalyzes the distribution of such knowledge. Future studies documenting the prevalence of CM/SM may provide more information regarding these findings. 

In this study, we did not identify age or weight differences between CM normal vs. abnormal and SM normal vs. abnormal dogs. In our previous study on a larger study population of Pomeranians [18], we found that older dogs were more likely to be CM/SM abnormal and SM abnormal dogs had a lower body weight than SM normal dogs. Our inability to replicate these findings in this smaller study population may represent a false negative finding. Alternatively, the finding in the other study might have been a false positive one. Future studies should evaluate for an association between weight and age and CM/SM status.

### 4.2. Associations between Imaging Assessments and CM/SM Status

The inter- and intra-observer reliability and agreement percentages must be taken into account when discussing the value of the measurements under investigation. The reader is referred to Part I of this study for discussion of the results thereof. 

#### 4.2.1. MRI-Based Qualitative Parameters

None of the studied parameters were significantly associated with SM classification as normal vs. abnormal based on odds ratios. These specific qualitative parameters, therefore, do not seem to be major indicators of predilection to SM or major consistent features of CM. While previous studies have looked at CSF-related parameters, including dynamic (flow) factors, the presence or absence of a CSF signal at these sites has not been linked to CM/SM before either [6,19,20].

#### 4.2.2. Quantitative MRI- and CT-Based Measurements

We assessed for significant differences between CM and SM classification groups (normal vs. abnormal) for both MRI- and CT-based measurements and found: 

(1) A significantly larger foramen magnum (i.e., dorsoventral height on midsagittal images) for both CM and SM abnormal dogs compared to CM and SM normal dogs based on MRI measurements, but not based on CT measurements.

Abnormal foramen magnum shape (including increased height) has been reported based on MRI, CT, and post-mortem findings in dogs with CM [1,21], and is commonly seen in various small- and toy-breed dogs [10]. Two studies found no difference in foramen magnum height for CKCS with or without SM [1,22]. In another MRI-based study with a single trained observer, no significant differences in foramen magnum height were found between dogs of various breeds with or without SM [23]. Longitudinal studies found that foramen magnum height increases over time in CKCS (with SM) based on MRI measurements [24,25]. 

When we take into account the results of the interobserver and intermodality reliability described in Part I of this study, caution is advised in putting too much weight on this observation on MRI in Pomeranians. Also, as foramen magnum dimensions may change over time [24], the measurement will not only differ between dogs, but also within dogs scanned at different times. These changes may in fact be beneficial. As CSF dynamics are influenced by CM and play a role in SM pathogenesis, a larger bony foramen magnum may positively influence CSF flow and lead to changes in both CM and SM. Indeed, cranioplastic surgical techniques such as ‘foramen magnum decompression’ center on the removal of bone and enlargement of the foramen magnum in order to improve CSF flow [26,27,28,29]. 

In reality, it must be remembered that a loss of bone (either not there at all or disappearing over time) does not mean that that space is empty. In dogs with dysplastic supraoccipital bones, post-mortem studies have revealed thin membranous structures in place of where bone would have been [21,30]. Although there are no specific studies on the properties of this in dogs, it is obviously a more dynamic structure (i.e., able to move under pressure changes) than bone [21]. When foramen magnum decompression is performed in dogs, the bony defect is often covered with either a (dural)graft or titanium mesh [26,27,28,29]. The long-term impacts of such cranioplastic techniques are not well studied. In any case, the dynamics at the level of the foramen magnum are not solely influenced by its bony borders. An anatomical bony foramen magnum is not always the same as the ‘functional foramen magnum’ which may be lined not only by bone, but also by membranous structures. 

Differentiation between membranous structures and bone would be best accomplished by combined CT and MRI studies, and confirmed by post-mortem studies. The fact that we found significant differences in foramen magnum height for CM/SM abnormal dogs compared to CM/SM normal dogs in MRI but not CT may reflect that while the bony foramen magnum size does not differ significantly, the foramen magnum size including membranous structures does differ. As MRI measurements are likely take into account membranous structures as well (as these structures, like bone, appear hypointense), the measurements will differ from those performed on CT images. Future studies might include comparison of CT and MRI for the identification of these structures, as well as their ability to complement each other in this matter. 

(2) A significantly greater distance between the cranial tip of dorsal arch of the atlas and the foramen magnum line based on MRI measurements, but not based on CT measurements for CM abnormal dogs compared to CM normal dogs. 

Conversely, we found a significantly shorter distance between the cranial tip of dorsal arch of the atlas and the foramen magnum line based on CT measurements, but not based on MRI measurements for SM abnormal vs. SM normal dogs.

When taking this measurement as an objective criterium for the diagnosis of atlanto-occipital overlapping, the was no indication of association between the presence of overlapping and CM. Cerda-Gonzalez et al. found an association between atlanto-occipital overlapping and cerebellar indentation or herniation (a feature of CM2) in a population of various dog breeds [23]. Another study using MRI and CT found that atlanto-occipital overlapping was less common in CKCS dogs (almost all of which have CM) than other breeds, and more likely in cases with a higher cerebellar compression ratio [12]. However, this ratio does not account for concurrent conditions or differentiate between conditions that both may cause cerebellar compression, such as CM and atlanto-occipital overlapping. As both CM and atlanto-occipital overlapping may cause cerebellar compression, these disorders may confuse observers and may be one explanation for interobserver disagreement on CM classification. 

The association of a shorter distance between the cranial tip of dorsal arch of the atlas and the foramen magnum line based on CT measurements with the presence of SM is in line with findings of a previous study on a population of various dog breeds, but in contrast to findings of other studies on various breeds and Chihuahuas [3,4,23,31,32]. A review on diagnostic imaging characteristics of CM/SM also mentions that the risk of SM increases with decreased atlanto-occipital distance [6], but the references listed there do not all entirely reflect this statement.

Taking into account the results of the interobserver and intermodality reliability for MRI and CT findings described in Part I of this study, and the discrepancy between studies in other breeds, these findings remain of questionable relevance. The differential findings for CT- and MRI-based measurements likely also reflect the differences in determination of the borders of the foramen magnum as discussed above for foramen magnum height measurements. Future studies evaluating the prevalence of atlanto-occipital overlapping in the Pomeranian are needed to assess the relationship with CM/SM and the value of such measurements.

(3) A significantly shorter clivus length based on both MRI and CT measurements for SM abnormal vs. normal dogs.

‘Clivus’ is a term mostly used in human medical literature, representing part of the basisphenoid and the whole basioccipital bone length. In veterinary literature, the clivus length is also referred to as ‘caudal fossa length’ or described as the distance between the dorsum sella turcica and the ventral margin of the foramen magnum [8,22]. Having a shorter basioccipital bone (major component of the clivus) is one of the features of brachycephalism [33,34]. Pomeranians are regarded as brachycephalic dogs [35]. Within this brachycephalic study population, the degree of brachycephalism based on the length of the basioccipital bone differs between SM abnormal vs. SM normal dogs. In other words, more ‘severe’ brachycephalism appears to predispose to development of SM in the Pomeranian. 

The finding of a shorter clivus length in dogs with SM compared to dogs without SM seems to be a robust finding due to the significance of the findings for both the modalities separately, despite only moderate intermodality agreement. This means that while measurements based on MRI may differ from those obtained on CT images, SM abnormal dogs still have a shorter clivus compared to SM normal dogs whether measured on MRI or CT. 

A shorter clivus length has been reported for people with Chiari malformation type I and SM [36]. However, a recent meta-analysis and systematic review did not identify clivus length as associated with SM [37]. 

A shorter basioccipital bone was reported in the CKCS in one study, but clivus length was not different between groups in two other studies [8,22,33]. Early closure times for the spheno-occipital synchondrosis have been linked to development of the abnormal skull phenotype of the CKCS compared to other dogs (brachycephalic and mesaticephalic dogs) [38]. Closure times for the spheno-occipital synchondrosis have not been reported for the Pomeranian dog breed.

(4) A significantly smaller caudal cranial fossa area (Area 1) based on both MRI and CT measurements for SM abnormal vs. normal dogs.

This finding is in agreement with previous studies in other breeds which found reduced caudal cranial fossa areas, area ratios, or volumes in affected dogs [4,22]. A reduced volume of the caudal cranial fossa is regarded as a feature of CM (particularly in CKCS dogs), which in turn is associated with SM [6,39]. However, there are studies that did not find a difference in caudal cranial fossa area or volume between dogs with or without SM [1,8,24]. One of these studies did find that the volume of the parenchyma in the caudal cranial fossa was relatively bigger for CKCS with SM than CKCS without SM [40]. 

In the human medical literature, a link between Chiari malformation (various types) and a small posterior fossa is reported [41,42]. Specifically, one study identified that patients with CM type I, but without SM, did not have significantly smaller posterior fossa volumes, whereas patients with CM type I and SM had volumes significantly smaller than normal [42]. 

(5) A significantly smaller rostral and middle cranial fossa area (Area 2) based on CT measurements, but not based on MRI measurements for SM abnormal vs. normal dogs.

Overcrowding in the cranial fossa (rostral and middle parts) can lead to altered CSF dynamics and pressure on the cerebellum, and has been linked to CM/SM in other breeds [4,6]. No association between cranial volumes for rostral, middle, or caudal cranial fossa spaces and SM was found in a study on CKCS and brachycephalic dogs [43]. 

Taking findings based on CT measurements for the rostral, middle, and caudal cranial fossa together, dogs with SM had a significantly smaller total cranial fossa midsagittal area than dogs without SM. This could reflect that a smaller cranial vault increases the risk of SM. Future morphometric and volumetric studies on Pomeranians with and without SM may provide more robust information on this matter.

(6) A significantly greater Angle 2 based on MRI measurements, but not based on CT measurements for SM abnormal vs. normal dogs.

Angle 2, the angle between the line from caudal tip of the basioccipital bone to the cranial tip of the dens axis and the line from caudal tip of the basioccipital bone to the midpoint of the caudal endplate of the axis, was selected as a measurement parameter to reflect alignment of the caudal aspect of the basioccipital bone (or ventral margin of the foramen magnum) with the atlas and axis, taking into account dysplasia or displacement of the dens axis. While the measurements of Angle 1 and 2 are not validated as diagnostic for craniocervical junction disorders such as atlantoaxial subluxation/instability or abnormalities such as medullary kinking, these or similar angles have been studied in other breeds in relation to CM/SM before and we would expect them to differ between dogs with or without CM/SM if such craniocervical disorders or abnormalities were associated with CM/SM [4,30,33]. Earlier studies reported variable results for the associations between CM/SM and other craniocervical disorders such as atlantoaxial subluxation/instability or medullary kinking [3,7,23,43,44]. 

While we did find a difference between SM normal and abnormal dogs for Angle 2 based on MRI measurements, the same measurements based on CT images did not confirm this finding. Taking into account interobserver and intermodality reliability findings described in Part I of this study, we suspect this finding might represent a false positive result (type I error) and thus cannot definitively conclude an association between Angle 2 and SM status in Pomeranians. Future studies looking into associations between these measurements or specific diagnoses and CM/SM in Pomeranians may provide more information on these considerations.

### 4.3. Limitations

A relevant limitation, or consideration, is the fact that research has shown that particularly SM is a disorder that may change over time [24,25,45,46,47]. This means that dogs that were classified as SM normal at the point in time of the MRI and CT studies could have become SM abnormal at a later point. The measurements and assessments performed on these dogs would be misclassified in the group of SM normal, retrospectively. Another limitation is that, although the study population is large in comparison to some previous studies on other breeds, the limited number of included dogs may still have resulted in false negative results when assessing for associations between measurements and CM/SM status of Pomeranians. Finally, although dogs were carefully positioned for both imaging studies, we cannot exclude an influence of positioning on the results of the measurements performed in this two-part study (see also discussion of Part I).

## 5. Conclusions

Several craniocervical MRI- and/or CT-based measurements were associated with the presence of CM/SM in Pomeranians in this study, including a significantly shorter clivus length and smaller caudal cranial fossa area for dogs with SM. These findings add to those already described in other breeds, and provide further insight into factors that may play a role in the pathogenesis of CM/SM in Pomeranians. 

## Figures and Tables

**Figure 1 animals-14-01859-f001:**
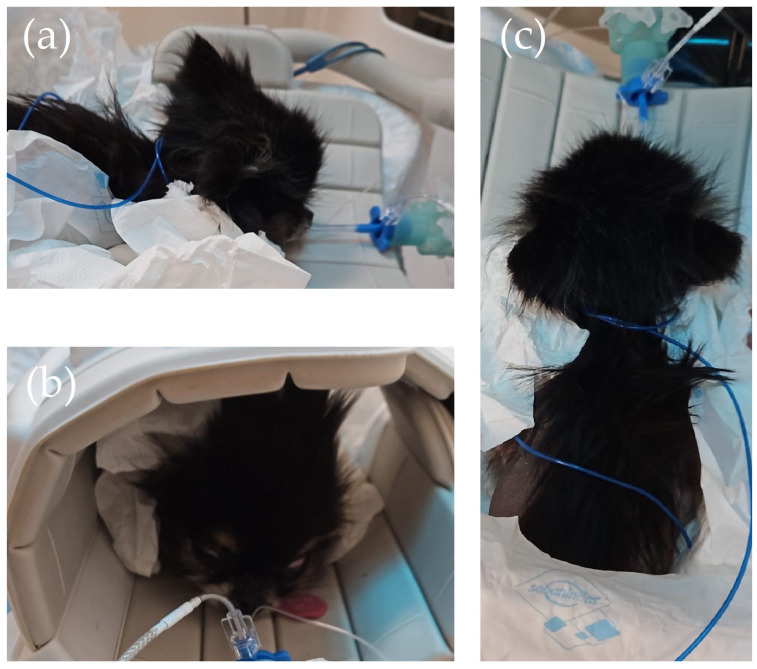
Positioning of the dogs for diagnostic imaging studies: (**a**) lateral view; (**b**) frontal view; and (**c**) dorsal view. For CT scans, a head rest substituted for the flexible coil (in the pictures) used for MRI scans.

**Table 1 animals-14-01859-t001:** Characteristics of the study population.

Total Study Population	99 (100%)
**Sex**	
Female	47 (47%) 44 intact, 3 neutered
Male	52 (53%) 46 intact, 6 neutered
**Age ^1^**	2.9 years (1.9–3.7)
**Weight ^1^**	3.2 kg (2.6–3.9)

^1^ Median and interquartile range.

**Table 2 animals-14-01859-t002:** Contingency table including numbers and percentages (of total) of included dogs’ CM and SM classifications.

Classification	SM Normal	SM Abnormal	Total
**CM normal**	20 (20%)	16 (16%)	36 (36%)
**CM abnormal**	29 (29%)	34 (34%)	63 (64%)
**Total**	49 (49%)	50 (51%)	99 (100%)

CM, Chiari-like malformation; SM, syringomyelia.

**Table 3 animals-14-01859-t003:** Odds ratios (95% confidence intervals) for qualitative MRI assessments of CM normal vs. abnormal classification groups.

MRI Qualitative Parameter	OR	CI 95%	*p*-Value
Presence of CSF signal between the cerebellum and the supraoccipitum	0.74	0.30–1.80	*p* = 0.51
Presence of CSF signal at the ventral aspect of the cervicomedullary junction	0.18	0.01–3.46	*p* = 0.27
Presence of CSF signal at the dorsal aspect of the cervicomedullary junction	1.90	0.62–5.80	*p* = 0.26

CI, confidence interval; CSF, cerebrospinal fluid; and OR, odds ratio.

**Table 4 animals-14-01859-t004:** Odds ratios (95% confidence intervals) for qualitative MRI assessments of SM normal vs. abnormal classification groups.

MRI Qualitative Parameter	OR	CI 95%	*p*-Value
Presence of CSF signal between the cerebellum and the supraoccipitum	1.07	0.45–2.55	*p* = 0.88
Presence of CSF signal at the ventral aspect of the cervicomedullary junction	0.14	0.01–2.51	*p* = 0.18
Presence of CSF signal at the dorsal aspect of the cervicomedullary junction	1.15	0.51–2.58	*p* = 0.74

CI, confidence interval; CSF, cerebrospinal fluid; and OR, odds ratio.

**Table 5 animals-14-01859-t005:** Differences between CM normal vs. abnormal classification groups for MRI- and CT-based measurements. Those items and values for which significant differences were found between groups are marked in bold.

Item	CM Normal	CM Abnormal	*p*-Value
**MRI**	**Mean (SD ^1^)**	**Mean (SD)**	
Distance between the os tentorium cerebelli and the dorsum sellae (cm)	2.61 (0.19)	2.62 (0.19)	0.8483
Length of the clivus (cm)	1.90 (0.14)	1.87 (0.16)	0.3623
**Height of the foramen magnum** **—‘foramen magnum line’ (cm)**	**1.75 (0.14)**	**1.83 (0.20)**	**0.0158**
**Distance between cranial tip of dorsal arch of the atlas and the foramen magnum line (mm)**	**0.00 (0.88)**	**0.55 (1.10)**	**0.0065**
Area 1 ^2^ (cm^2^)	4.42 (0.44)	4.31 (0.52)	0.2392
Area 2 ^3^ (cm^2^)	14.24 (0.96)	13.97 (1.11)	0.2127
Angle 1 ^4^ (degrees)	119.10 (8.40)	118.91 (9.87)	0.9195
Angle 2 ^5^ (degrees)	26.92 (6.38)	28.21 (9.98)	0.4354
**CT**			
Distance between the os tentorium cerebelli and the dorsum sellae (cm)	2.49 (0.30)	2.57 (0.23)	0.1600
Length of the clivus (cm)	1.92 (0.19)	1.84 (0.20)	0.0673
Height of the foramen magnum —‘foramen magnum line’ (cm)	1.79 (0.19)	1.77 (0.20)	0.7547
Distance between cranial tip of dorsal arch of the atlas and the foramen magnum line (mm)	2.12 (1.43)	1.97 (1.57)	0.6194
Area 1 ^2^ (cm^2^)	5.24 (0.63)	5.11 (0.68)	0.3461
Area 2 ^3^ (cm^2^)	15.08 (1.09)	14.71 (1.23)	0.1319
Angle 1 ^4^ (degrees)	104.34 (13.70)	107.27 (13.79)	0.3090
Angle 2 ^5^ (degrees)	21.51 (9.71)	23.57 (10.57)	0.3291

^1^ Standard deviation, ^2^ Area 1 = area between the yellow line and the red line and osseous structures (caudal cranial fossa area, Figure 2), ^3^ Area 2 = area rostral to the yellow line (rostral and middle cranial fossa area, Figure 2), ^4^ Angle 1 = angle between the line from caudal tip of the basioccipital bone to the cranial tip of the dens axis (red line) and the line from the ventral aspect of the supraoccipital bone to the cranial tip of the dens axis (green line) (Figure 3), and ^5^ Angle 2 = angle between the line from caudal tip of the basioccipital bone to the cranial tip of the dens axis (red line) and the line from caudal tip of the basioccipital bone to the midpoint of the caudal endplate of the axis (blue line) (Figure 3).

**Table 6 animals-14-01859-t006:** Differences between SM normal vs. abnormal classification groups for MRI- and CT-based measurements. Those items and values for which significant differences were found between groups are marked in bold.

Item	SM Normal	SM Abnormal	*p*-Value
**MRI**	**Mean (SD ^1^)**	**Mean (SD)**	
Distance between the os tentorium cerebelli and the dorsum sellae (cm)	2.63 (0.20)	2.60 (0.18)	0.5329
**Length of the clivus (cm)**	**1.93 (0.13)**	**1.84 (0.16)**	**0.0063**
**Height of the foramen magnum** **—‘foramen magnum line’ (cm)**	**1.75 (0.12)**	**1.85 (0.22)**	**0.0073**
Distance between cranial tip of dorsal arch of the atlas and the foramen magnum line (mm)	0.34 (1.04)	0.36 (1.07)	0.9231
**Area 1 ^2^ (cm^2^)**	**4.46 (0.49)**	**4.23 (0.47)**	**0.0230**
Area 2 ^3^ (cm^2^)	14.20 (0.93)	13.93 (1.16)	0.2004
Angle 1 ^4^ (degrees)	117.81 (9.31)	120.13 (9.28)	0.2185
**Angle 2 ^5^ (degrees)**	**25.68 (6.64)**	**29.76 (10.21)**	**0.0207**
**CT**			
Distance between the os tentorium cerebelli and the dorsum sellae (cm)	2.55 (0.22)	2.54 (0.29)	0.8726
**Length of the clivus (cm)**	**1.92 (0.16)**	**1.81 (0.22)**	**0.0057**
Height of the foramen magnum —‘foramen magnum line’ (cm)	1.78 (0.19)	1.78 (0.20)	0.9094
**Distance between cranial tip of dorsal arch of the atlas and the foramen magnum line (mm)**	**2.34 (1.57)**	**1.72 (1.42)**	**0.0402**
**Area 1 ^2^ (cm^2^)**	**5.30 (0.68)**	**5.01 (0.61)**	**0.0239**
**Area 2 ^3^ (cm^2^)**	**15.09 (1.06)**	**14.61 (1.27)**	**0.0455**
Angle 1 ^4^ (degrees)	105.39 (13.70)	106.99 (13.91)	0.5672
Angle 2 ^5^ (degrees)	22.27 (10.23)	23.36 (10.37)	0.5990

^1^ Standard deviation, ^2^ Area 1 = area between the yellow line and the red line and osseous structures (caudal cranial fossa area, Figure 2), ^3^ Area 2 = area rostral to the yellow line (rostral and middle cranial fossa area, Figure 2), ^4^ Angle 1 = angle between the line from caudal tip of the basioccipital bone to the cranial tip of the dens axis (red line) and the line from the ventral aspect of the supraoccipital bone to the cranial tip of the dens axis (green line) (Figure 3), and ^5^ Angle 2 = angle between the line from caudal tip of the basioccipital bone to the cranial tip of the dens axis (red line) and the line from caudal tip of the basioccipital bone to the midpoint of the caudal endplate of the axis (blue line) (Figure 3).

## Data Availability

The raw data supporting the conclusions of this article will be made available by the authors on request.

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
