# Peer review of "Craniocervical Morphometry in Pomeranians—Part II: Associations with Chiari-like Malformation and Syringomyelia"

_animals, 2024, doi:10.3390/ani14131859_

Round 1

Reviewer 1 Report

Comments and Suggestions for Authors

The reviewed article is the second part of a study on cranial and cervical spine morphometry in Pomeranian dogs suffering from CM/SM. The purpose of the article is to establish the relationship of CT and MRI measurements of cranial fossa and craniocervical junction in the study group.  The subject matter of the article seems important and timely, due to the large percentage of dogs affected by this condition, which significantly affects the quality of life of dogs and is also an important issue in the context of selection of individuals for breeding.  In the introduction, the authors briefly characterize CM/SM and refer to the results of the first part of the article.  In Materials and methods, the authors indicate precise exclusion and inclusion criteria. The authors also describe how measurements were made by placing references in the figures. The way the figures are labeled needs improvement, as the color-described lines are unreadable.  It is difficult to distinguish blue from green in the figures, as well as the lines are too thin.  The results are presented correctly and clearly. A correct statistical analysis was also performed.  The discussion discusses in detail and confronts the results obtained with the current literature, and provides a good basis for correct conclusions. Citations are up-to-date, extensive and correct, with no inappropriate self-citation. In conclusion, the article, after minor corrections ( concerning the markings of the photos), is suitable for publication.

Author Response

Dear reviewer,

Our sincere thanks for your time and effort spent to review this manuscript.
We have addressed your comment concerning the lines in the figures. Thank you for pointing this out to us.

Kind regards,

Authors

Reviewer 2 Report

Comments and Suggestions for Authors

Dear Aothors

I have been fortunate to have read Part I and II. Even though the research is good the papers are very similar. Could there not have been just one paper?

Author Response

Dear reviewer,

Our sincere thanks for your time and effort spent to review this manuscript.

Indeed, it is a two-part study / publication. We originally included both parts in one large manuscript. However, we received feedback and discussed amongst co-authors that this was confusingly long and contained many data. The results and discussion sections became difficult to read since we had to go back and forth between the results of part I and part II to take them into account for each, respectively. Therefore, we decided and discussed with the editors of the journal if we could/should submit two parts at the same time. Overall, this was preferred for the benefit of the reader.

We hope this explains our train of thought and would be glad to hear if this is satisfactory to you.

Kind regards,

Authors

Reviewer 3 Report

Comments and Suggestions for Authors

Dear Authors

I reviewed the manuscript entitled "Craniocervical morphometry in Pomeranians – Part II: Associations with Chiari-like malformation and syringomyelia" . This part of the manuscript analyzes the association between the morphometry of the skull and craniocervical region and the presence of Chiari malformation and syringomyelia in Pomeranians. All dogs were evaluated with CT and MRI. Similarly to part I, also this part of the manuscript is well written and the topic is interesting, since it points out significant association between specific morphometry of the skull and CM/SM. I have only minor comments, therefore I recommend minor revision

Specific comments

Materials and methods (See part I)

line 92-93 Dogs were  positioned in sternal recumbency on the horizontal surface of the table with the head in a flexible coil (MRI) or a head rest (CT). Please describe more thoroughly how dogs were positioned, in order to reproduce the measurements (for example which kind of head rest, etc). Perhaps an image would be helpful.

line 103-104: For the CT scans, 0.6 and 0.8 mm slices were obtained with 130 kV and 80 mAs/slice and 130 kV and 220 mAs/slice and reconstructed in dorsal and sagittal  planes and 3D for further analysis. Please expand CT protocol. Rotation tube ? Pitch? Algorithm? 

Did the Authors try to evaluate SM also in CT images with HU measurements?

Were the MRI measurements performed only in T2W images?

Discussion

376-379 "The fact that we found significant differences in foramen magnum height for CM/SM abnormal dogs compared to CM/SM normal dogs on MRI but not CT may reflect that while the bony foramen magnum size does not differ significantly, the foramen magnum size including membranous structures does differ." Did the Authors try to measure the foramen magnum on CT images using soft tissue windows? Did the Authors reviewed the CT images with the MRI images as reference, to see if membranous structures can be visualized in CT?

Author Response

Dear reviewer,

Our sincere thanks for your time and effort spent to review this manuscript.

For line 92-93: We have included a new figure (photographs) that shows the positioning of the dogs.

For line 103-104: We have expanded the information on the CT protocol.

We did not evaluate SM on CT images with HU measurements (we decided not to as MRI is the gold standard and a previous study already compared MRI versus CT - https://pubmed.ncbi.nlm.nih.gov/32329949/).

All MRI sequences included in the methods were available for the observers and observers were allowed to perform measurements on the images side by side to facilitate the most accurate measurements (exact placement, of course, left to the observer in question). We have included an additional sentence in section 2.1. to clarify this.

For lines 376-379:

We did not include soft tissues in the measurements on CT images – we aimed to include the bony margins of the foramen magnum on both CT and MRI. However, for the MRI evaluation, it may be that we did (inadvertently) include ‘soft tissues’ (membranous structures).
We do agree that it would be interesting to investigate if CT and MRI can both be used or can compliment each other to visualize those structures. This was not included in our methodology, but may be included in future studies. We have included a comment on this in the new version’s discussion at this section. Thank you for your input.

Kind regards,

Authors

Round 2

Reviewer 2 Report

Comments and Suggestions for Authors

Thank you for comments